



# What if extreme droughts occur more frequently? - Mechanisms and
# limits of forest adaptation in pine monocultures and mixed forests in
# Berlin-Brandenburg, Germany
Jamir Priesner[1,2], Boris Sakschewski[2], Maik Billing[2], Werner von Bloh[2], Sebastian Fiedler[1], Sarah
Bereswill[2], Kirsten Thonicke[2], Britta Tietjen[1,3]
[1]Freie Universität Berlin, Theoretical Ecology, Institute of Biology, Königin-Luise-Straße 2/4, Gartenhaus, 14195 Berlin,
Germany
[2]Potsdam Institute for Climate Impact Research, Member of the Leibniz Association, Telegraphenberg A31, 14473 Potsdam,
Germany
[3]Berlin-Brandenburg Institute of Advanced Biodiversity Research (BBIB), 14195 Berlin, Germany
*Correspondence to*: Jamir Priesner (jamir.priesner@fu-berlin.de)
**Abstract.** Forests in Eastern Germany are already experiencing the detrimental effects of droughts, exemplified by the
severe conditions of the 2018 drought year. With climate change, such extreme events are expected to become more frequent
and severe. Previous work suggests that mixed forests exhibit greater resilience against droughts than monocultures. Our
study aims to investigate the impact of increased frequency of extreme droughts, such as those seen in 2018, on biomass,
structure and traits of forests in the Eastern German federal states of Berlin and Brandenburg.
Utilizing the flexible-trait Dynamic Global Vegetation Model LPJmL-FIT, we simulate the growth and competition of
individual trees in both, pine monoculture forest and mixed forest. The trees belong to different plant functional types or in
case of pine forest are parametrized as *Pinus sylvestris*. We create drought scenarios from high resolution climate input data
by re-shuffling the contemporary climate with increased frequencies of the extreme drought year 2018. For each scenario, we
simulated vegetation dynamics over 800 simulation years which allowed us to analyze shorter-term impacts, in the first
decades of the drought scenarios, as well as the long-term adaptation of the two forest types to those new climate normals.
We evaluated the resulting long-term changes in biomass, plant functional traits and forest structure to examine the new
equilibrium state emerging for each scenario.
Our findings revealed nuanced responses to increased drought frequency. In pine monoculture forests, increased drought
frequency reduced biomass and increased biomass variance, indicating higher system instability. Conversely, in mixed
forests, biomass initially declined in scenarios with increased drought frequency but eventually recovered and even exceeded
baseline levels after 100-150 years. We explain recovery and increase of biomass through two forest adaptation mechanisms;
first, we saw a shift in the plant community towards broadleaved trees and second, plant traits shifted towards increased
average wood density, decreased average tree height and increased average tree age. However, for the most extreme scenario
with drought occurring each year, the adaptive capacity of the mixed forest was exceeded and the biomass halved compared
to the baseline scenario. In our study, for the first time LPJmL-FIT is used with a resolution as high as 2 by 2 km², which
allows us to observe spatial heterogeneity drought impacts within the Berlin-Brandenburg area. Pine monocultures suffered,
especially in the warmer urban areas and mixed forests in the central-west of Brandenburg, benefitted in the long term.





This study highlights the capacity of natural mixed forests in contrast to pine monocultures to adapt to increasing drought
frequency up to a certain limit. The results underscore the importance of considering biodiversity in forest management
strategies, especially with regard to more frequent dry periods under climate change.

## 1 Introduction


With temperatures rising at about twice the global average rate, Europe is the fastest warming continent (Copernicus 2023)
and is exposed to more intense and frequent climate extremes (Pradhan et al., 2022; Treydte et al., 2024). Within Germany,
Brandenburg is one of the driest regions (Dittmann et al., 2024), with a  warming of 1.1°C over the last decades (LfU
Brandenburg, 2022) and consequently increasing evaporation and decreasing groundwater recharge (LfU Brandenburg,
2022). Increasing drought severity and frequency have been observed to damage forests in Brandenburg and to increase the
severity of wildfires (Land Brandenburg, 2023; LfU Brandenburg, 2021).
Germany was heavily affected by two consecutive drought years in 2018-2019, where the annual precipitation amount was
so low, that drought effects extended into 2020 (Büntgen et al., 2021). The extremely dry and hot growing season (March -
November) in 2018 was record-breaking both in its high temperatures and low precipitation (Zscheischler and Fischer, 2020)
and led to a record in burned forest area due to wildfires in Brandenburg (1664 ha burned area in Brandenburg,
Landeskompetenzzentrum Forst Eberswalde 2018). This compound event affected agricultural production requiring federal
states to warrant aid payments of 340 million Euros to farmers with at least 30 percent of yield loss (Reinermann et al. 2019;
Buras et al. 2020). During the consecutive drought years the fraction of trees with signs of damage increased from 53 % in
2017 to 92% in 2022 (Land Brandenburg, 2023). In 2021, 26% of the forest area covered with pine in Brandenburg (Berlin:
20%) showed significant damage in their tree crowns while the area affected in Brandenburg's beech and oak forest
amounted to 40 and 42%, respectively (Bundesministerium für Ernährung und Landwirtschaft, 2021).
Since medieval times, natural forests have been cleared for agriculture, pastures, with natural or semi-natural forests left in
only a few small regions in Europe (Barredo et al., 2021; Bengtsson et al., 2000; Lamentowicz et al., 2020). In Europe,
systematic forest management roots back into the 19th century (Niedertscheider et al., 2014). Today, most European forests
are highly managed, they are often planted single-species monocultures to provide people with timber from high productive
forests. In Brandenburg, pine trees make up 70.1 % of the forest area, followed by oak and beech with only 6.7 % and 3.3 %,
respectively (Land Brandenburg, 2023). To act against increasing forest losses from climate extremes and to support forest
adaptation to a changing climate, increasing biodiversity has been suggested as one solution. According to the biological
insurance hypothesis, ecosystems with high biodiversity can better maintain ecosystem functioning under external pressure.
In addition, diverse forests can hold a larger portfolio of plant strategies that can help them to adapt to the new
environmental conditions. Monocultural ecosystems, however, lack the required response diversity to maintain ecosystem
functioning under changing environmental conditions (Mori et al., 2013; Yachi and Loreau, 1999).
Changing climate conditions can lead to environmental filtering and thus to a shift in the spatial domain where species can
occur and be productive. Respective shifts in species' spatial distribution are among the most significant and most widely
discussed ways of how forests in the northern hemisphere react/adapt to climate change (Astigarraga et al., 2024; Fei et al.,
2017; Lenoir and Svenning, 2015; Parmesan and Yohe, 2003; Rabasa et al., 2013; Rubenstein et al., 2020, 2023). Due to the
increase in water deficit and temperature, range shifts upward and poleward are expected by ecological theory (Bonebrake et
al., 2018; Lenoir and Svenning, 2015). While there are observations of species for which these expected shifts happen (Chen
et al., 2011; Lenoir and Svenning, 2015; Parmesan and Yohe, 2003), many species show multiple directions in response to
climate change (Fei et al., 2017; Rabasa et al., 2013; Rubenstein et al., 2023; Zhu et al., 2014). In addition to a range shift in
species, changes can also occur in stocks that persist at a specific location, particularly with regard to productivity. However,





the effects of climate change on European forests remain unclear (Pretzsch et al., 2023). While generally there is a trend
toward more productivity temperate European forests (Charru et al., 2017; Pretzsch et al., 2014, 2023; Zhu et al., 2014),
increasing drought events interrupt this trend (Martinez del Castillo et al., 2022; Piovesan et al., 2008; Schmied et al., 2023;
Schuldt et al., 2020; Williams et al., 2013). Due to contrasting trends within regions (Galván et al., 2014; Pretzsch et al.,
2023) and among species (Martinez del Castillo et al., 2022; Pretzsch et al., 2014, 2020, 2023), understanding of long-term
shifts is still lacking.
Building on the knowledge that more diverse forest ecosystems could be more resilient, recent forestry programmes in
Germany for example aim at increasing deciduous tree cover to adapt forests to future climate change conditions (Land
Brandenburg, 2011; Wessely et al., 2024). It is supported by future projections of decadal, average changes in forest
dynamics and tree species distribution (e.g., Wessely et al. 2024) and how it affects forests to provide ecosystem services in
Germany (Gregor et al., 2022; Gutsch et al., 2018). Recent model applications studied the importance of functional diversity
for future forest adaptation (Billing et al. 2022, 2024). However, we still have a limited understanding on the mechanisms
and limits of diverse forests to adapt to an increasing frequency of climate extremes as the new climate normals.
In addition to biodiversity and species identity, it is also useful to consider structural and functional plant traits that
determine the reactions to environmental factors and their changes but can also influence the functioning of ecosystems
(Sterk et al., 2013; Suding et al., 2008). Wood density and specific leaf area might strongly impact species' responses to
climate change. Some studies suggest that higher wood density correlates with drier and warmer climate (Nabais et al. 2018;
S.-B. Zhang et al. 2011; Swenson and Enquist 2007; Nelson et al. 2020, Bouchard et al. 2024). Most notably, in a recent
global tree inventory analysis for temperate forests Bouchard et al. (2024) found higher wood density with decreasing
rainfall (below values of 1000 mm/a, which would also apply to climate conditions in Brandenburg). Fei et al. (2017)
observed that in the Eastern part of the United States of America tree species that shifted to drier areas had higher median
wood density. A global meta-analysis of tree mortality in response to drought found that in addition to wood density also
specific leaf area (SLA) explain drought responses, where trees having a lower SLA showed lower mortality responses
(Greenwood et al., 2017). Also experimental results show that individuals of different tree species from the Mediterranean
area growing under drought stress had a decreased SLA in comparison to individuals of the same species growing in the
control (Valladares and Sánchez-Gómez, 2006), showing that this might be a potential adaptation mechanism.
Because forests develop and change on decadal time scales, respective assessments of climate-extreme impacts on
biodiversity-ecosystem function relationships are difficult to conduct in a field experiment. Instead, biodiversity-enhanced,
process-based vegetation modeling can be applied to project and explain how climate extremes affect functional trait
composition and ecosystem function in diverse forests and compare them against the performance of monoculture forests.
However, climate models most likely underestimate the frequency of hot dry compound events like the 2018 drought
(Zscheischler and Fischer 2020; van der Wiel et al. 2021) that were much more rare in the past. As a result, vegetation
models using these data cannot accurately simulate the impact of increased drought frequency. To overcome this problem,
we take a simplistic approach of designing climate scenarios with artificially increased drought frequency for the area of
Berlin and Brandenburg in Germany. We use these artificial drought scenarios as input data for the flexible-individual trait
Dynamic Global Vegetation Model LPJmL-FIT (Sakschewski et al. 2015, Thonicke et al. 2020) that simulates functional
and structural trait changes in conjunction with ecosystem functions under varying climate and soil conditions. We then
analyze how in Brandenburg and Berlin temperate mixed forests and pine monoculture forest (parameterizing *Pinus*
*sylvestris* trees) perform and adapt to changing frequency of climate extremes. However, in both forest types, forest
management is not considered, which means that the pine monoculture forest can be regarded as a semi-natural forest.
Additionally, we assume that the entire study area is covered by forest to take advantage of the high-resolution climate data
and include urban forest areas. In this context, this study aims to answer the following questions:





1) Does a diverse natural forest have a higher resilience against an increased frequency of extreme drought years such as
2018 than a pine monoculture forest?
2) What are the underlying mechanisms that enable forests in Brandenburg to adapt to the increased frequency of extreme
droughts? In particular, how do these mechanisms manifest in the shifts in tree community composition, and changes in the
traits spectrum within individual plant functional types?
3) Is there spatial variability in the response of the two forest types towards droughts across Berlin and Brandenburg?
We first describe how biomass of the pine monoculture vs. temperate mixed forest is changing under the different drought
extreme scenarios, before we analyze how structural and functional traits explain the underlying mechanisms and how these
mechanisms differ between PFTs.
**2 Methods**
We created artificial climate data sets with increased drought frequencies using high-resolution climate data compiled for the
study area Berlin-Brandenburg as the baseline (Bart et al., under review). Our new drought scenarios contain weather data
from 1980-2022, to which we have added the drought year 2018 with varying frequency. We investigated the impact that
these scenarios might have on pine monocultures which currently dominate managed forests in the study area and on mixed
forest as its natural analogue. We applied the flexible individual traits Dynamic Global Vegetation Model LPJmL-FIT to two
plant community configurations, i) a pine monoculture forest and ii) a mixed forest and calculated resulting forest
development for a baseline scenario (the original climate data set) and to our new drought scenarios (see below). We
simulate the study area to be fully covered by vegetation, neglecting land used for settlements and agriculture. Forest
management, such as thinning or logging, was not simulated in any of the configurations. We then analyzed changes in
vegetation dynamics and in plant characteristics at the centennial time scale to analyze the short- and long-term ability of
forests to adapt to an increased frequency of extreme droughts.
**2.1 Model description: The flexible-trait DGVM LPJmL-FIT**
The dynamic flexible-trait vegetation model LPJmL-FIT ('Lund-Potsdam-Jena managed Land – Flexible Individual Traits')
is a process-based Dynamic Global Vegetation Model (DGVM). It simulates the establishment, growth, competition and
mortality of individual trees using a forest gap approach. Tree individuals can differ in their functional traits according to the
leaf and stem economics spectrum (Sakschewski et al., 2015; Thonicke et al., 2020). The spatial resolution of model
simulations depend on the resolution of the input data. For each grid cell, the model requires soil texture as well as daily
climate input data (temperature, precipitation, and radiation) and atmospheric $CO_2$ concentration to calculate soil hydrology
and vegetation dynamics. Grid cells are further subdivided into independent forest patches of 10 m by 10 m on which tree
individuals compete for water and light. The present study uses the model version as described in Thonicke et al. (2020) and
Billing et al. (2024) and has been extensively validated. In addition, we adopted the variable rooting scheme described in
Sakschewski et al. (2020) to allow for diverse tree rooting strategies and excluded gras PFTs from our simulations.
Tree individuals are typically categorized into broad Plant Functional Types (PFTs) representing main ecological
characteristics of natural vegetation at the biome level as in the standard model LPJmL (Schaphoff et al. 2018). However, the
model can also be parameterized for specific species. In LPJmL-FIT newly established tree individuals are randomly
assigned to PFTs, if there is more than one PFT simulated at the same time. Key functional traits, such as specific leaf area
(SLA) and wood density (WD), are then randomly sampled out of the PFT- or species-specific ranges and remain constant





over a tree's life. Other functional traits (e.g. leaf nitrogen content) are connected to SLA and WD via trade-offs according to
the plant economics spectrum. Trees compete for light and water in independent 10 m by 10 m forest patches. Their crown
area and leaf area index control their capacity to absorb photosynthetic active radiation. Water uptake depends on root depth
and soil moisture availability. The amount of absorbed photosynthetic active radiation, soil water uptake and other
environmental factors such as temperature and atmospheric $CO_2$ concentration determine the gross primary production
(GPP) via the process of photosynthesis. Autotrophic respiration is divided into maintenance and growth respiration, both of
which are temperature-dependent and linked to the tree's biomass and GPP. Carbon that is lost through autotrophic
respiration is subtracted from GPP, resulting in net primary productivity (NPP), which represents the carbon available for
new growth. The allocation of NPP to various parts of each individual tree—roots, stems and leaves—is modeled based on
the specific strategies of each PFT to optimize resource use in different environmental conditions (Schaphoff et al., 2018).
Over time, performance and competition determine tree survival and growth. Via these processes, climate, soil properties
and competition conditions filter locally best adapted (environmental filtering) and best performing (competitive filtering)
tree individuals. That is, LPJmL-FIT can simulate functionally diverse forests but also monocultures that would grow under
the sole influence of climate and soil conditions. An illustrative video of forest community assembly is available in Billing et
al. (2024), Video 1, and can be found under the following link: https://www.pik-
potsdam.de/~billing/video/2023/spinup_LPJmLFIT.mp4. In this animation, each tree is colored according to its SLA or WD
value assigned at establishment.

## 2.2 Data and simulation experiments

In our simulations the area of Brandenburg and Berlin was represented by a grid of 7073 cells with ~2 x 2 km resolution. For
each grid cell, 80 patches of 10 m x 10 m patch size were simulated, representing the forest of the total grid cell. We run the
model with climate data derived from the Central Europe Refined analysis version 2 (CER v2) (Bart et al., under review).
This dataset was generated by dynamical downscaling of ERA5 reanalysis forcing data provided by the European Centre for
Medium-Range Weather Forecasts (ECMWF) for the area of Berlin and Brandenburg, utilizing the Weather Research and
Forecasting (WRF) model version 4.3.3. The climate data covers the period from 1980 to 2022 with daily temporal
resolution and 2 x 2 km spatial resolution. We first created a climate set for a 1000-year model spin up, randomly drawing
from the climate input years 1980-2022. To assess the effects of droughts, we afterwards manipulated the original climate
dataset by artificially adding the drought year 2018 to the data in increasing frequencies. For a slightly wetter scenario
(Scenario A), we only used data from 1980-2001 (i.e. frequency of drought year 2018 = 0), and for the baseline scenario
(Scenario B, frequency = 0.02), we used the original full dataset from 1980-2022 without any manipulation. Five additional
scenarios contained the years 2001-2022 plus the drought year 2018 at varying frequencies (Scenario C: 0.05 to Scenario G:
0.68). For the last scenario we only took data from the year 2018 (Scenario H: 1.0). Increasing the frequency of the 2018
drought year also changed the mean climate. To quantify the drought effect, we calculated the mean annual Maximum
Climatic Water Deficit (MCWD) following (Sakschewski et al., 2021). The absolute values and their deviation from the
baseline climate as well as the frequency the year 2018 for each scenario are shown in Table 1. To create the full weather
data sets to run the model for different drought scenarios (Scenarios A-H), we randomly draw 800 years from the respective
manipulated climate dataset (Figure 1).
**Table 1:** *Characteristics of drought scenarios. Frequency of the year 2018 in manipulated data and resulting mean*
*maximum climatic water deficit (MCWD) and mean temperature (T) and deviation from baseline (Scenario B) for each of*
*the scenarios.*

| Scenario | Frequency of Year 2018 [year⁻¹] | MCWD [mm] | Δ MCWD [mm] | T [Celsius] | ΔT [Celsius] |
|---|---|---|---|---|---|
| **A: 1980-2001** | 0 | -328.7 | 36.0 | 8.9 | -0.42 |





| | | | | | |
|---|---|---|---|---|---|
| **B: 1980-2022 (Baseline)** | 0.02 | -364.6 | 0 | 9.31 | 0 |
| **C: 2001-2022** | 0.05 | -400.3 | -35.7 | 9.71 | 0.4 |
| **D: 2001-2022+7x2018** | 0.28 | -455.2 | -90.6 | 9.94 | 0.63 |
| **E: 2001-2022+14x2018** | 0.42 | -492.3 | -127.7 | 10.08 | 0.77 |
| **F: 2001-2022+22x2018** | 0.52 | -521.1 | -156.5 | 10.19 | 0.88 |
| **G: 2001-2022+44x2018** | 0.68 | -560.2 | -195.5 | 10.35 | 1.04 |
| **H: 2018 only** | 1 | -632.5 | -267.9 | 10.67 | 1.36 |


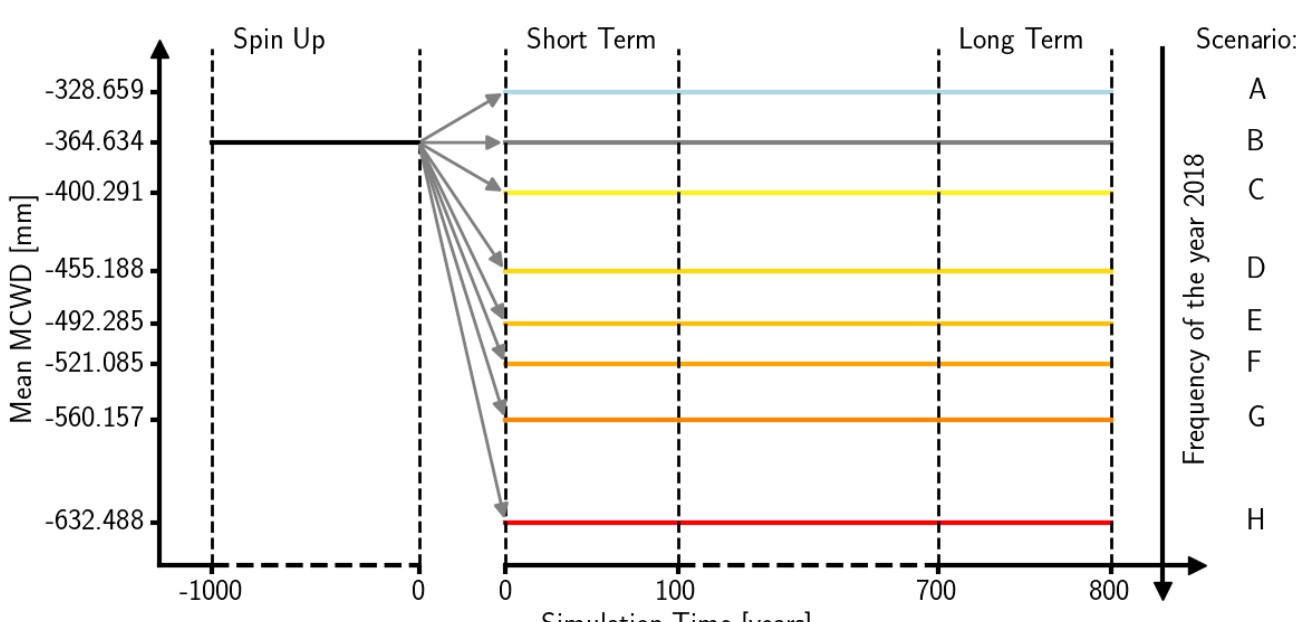


**Figure 1:** *Scheme of the simulation protocol. After 1000 years of spin up with shuffled Central Europe Refined analysis*
*version 2 (CER v2) data from 1980 - 2022, 800 years of Scenarios A-H with decreasing mean Maximum Water Deficit*
*(MCWD) and increasing frequency of the year 2018 follow. The frequency of the year 2018 was increased by adding an*
*increasing number of the year 2018 to the pool from which each year was drawn (see Table 1 for details).*

Soil depth data was sourced from Pelletier et al. (2016) and subsequently re-gridded from an original resolution of about 1
km to match the climate-data grid resolution. Soil type information was obtained from the Harmonized World Soil Database
(HWSD) (FAO and IIASA, 2023) and aggregated to match the grid cell resolution using the LandInG package (Ostberg et
al., 2023).



We ran the model for two forest configurations, a monoculture including only trees of *Pinus sylvestris* (pine monoculture,
hereafter) and a diverse temperate mixed forest (mixed forest, hereafter). The temperate mixed forest included four major
PFTs; temperate broad-leaved summergreen (T-BL), temperate needle-leaved evergreen (T-NL), boreal needle-leaved
evergreen (B-NL), and boreal broad-leaved summergreen (B-BL). In the mixed forest configuration, any PFT can be
established in any forest patch at any time, following the approach used in Thonicke et al. (2020), with the establishment rate
of new trees depending on light availability on the forest floor. As described above, their key functional traits are randomly
sampled out of the PFT-specific ranges. These are obtained from the TRY database (Kattge et al., 2011) as described by
Sakschewski et al. (2015). For the pine monoculture, only pine trees can be established, which were parameterized using the
boreal needle-leaved PFT and restricting the ranges for SLA and WD to the 25th and 75th percentile of their respective
distributions in the TRY database (Kattge et al., 2011) for *Pinus sylvestris*. SLA and WD ranges of all PFTs and *Pinus*
*sylvestris* are provided in Appendix A, Table A1. Other important differences between the tree types are their temperature
limits for establishment which reflects chilling requirements and frost tolerance as well as their optimum temperature range
for photosynthesis (see Appendix A, Table A2).
The model spin up started with the establishment of saplings on bare ground (illustrated in Billing et al. 2024, Video 1) and
was run for 1000 years of simulation for each forest configuration with the spin up climate dataset. Afterwards, we ran the
model for 800 years with the different drought scenarios for each forest configuration (Figure 1).

**2.3 Evaluation of simulation outcomes**

We evaluated the overall resilience of both pine monoculture and mixed forests against an increased frequency of extreme
droughts by calculating the mean for the  above- and belowground biomass (kgC/m²) across the entire study area for each
year. We compared the short-term (years 1 - 100) and the equilibrium (long-term, years 701-800) biomass of each scenario
with the baseline scenario (Scenario B). Then we analyzed different adaptation mechanisms to increased drought frequencies
by calculating the mean above - and belowground biomass [kgC/m²], the mean number of trees per m² (tree density), the
mean tree height [m], the mean tree wood density [kg/m³] and the mean tree age [years] over the e study region for the last
hundred years of the simulation for each scenario and forest configuration. We further investigated how these mechanisms
manifested in the mixed forest including changes in the tree community composition and changes in traits for all tree
individuals belonging to a particular PFT. For this, we calculate the mean tree height [m], tree wood density [kg/m³], tree age
[years], SLA [m²/g], mortality [probability/year] and mean growth speed [years] for each PFT during the last 100 simulation
years of each experiment. The growth speed of an individual tree was defined as the time to reach a height of 15 meters.
Trees that did not reach this size were not evaluated. For the calculation of the means for tree density, height, wood density,
SLA, mortality and age, trees were weighted with their biomass and trees smaller than 5 m height were excluded to prevent
an evaluation bias towards the multiple small trees.
To assess spatial heterogeneity in forest responses, we computed the long-term (i.e. mean over the last 100 simulation years)
impact of our drought scenarios on the spatial biomass variation of the pine monoculture and the mixed forest configuration
across Berlin and Brandenburg.

**3 Results**

The overall resilience in the pine monoculture and mixed forests against a higher frequency of droughts was assessed by
calculating mean biomass over the entire simulation domain, assuming the Berlin-Brandenburg area is covered by vegetation
only, for each simulated year and scenario (Fig. 2). While the wet scenario (Scenario A, without the 2018 drought year) and
the baseline scenario B show stable biomass in the pine monoculture forest over the entire simulation period, all drought



scenarios lead to biomass loss and biomass remains lower than under the Scenarios A and B (Fig. 2a). In the mixed forest,
biomass increased again after an initial phase of biomass decline of 50-150 years (Fig. 2b). Exceptions are the wet Scenario
A and the most extreme climate Scenario H, where biomass declines and remains at a lower level (Fig. 2b). After an initial
phase of decrease in biomass in both forest configurations, the biomass stabilized under all drought scenarios, fluctuating
around a new stable state.

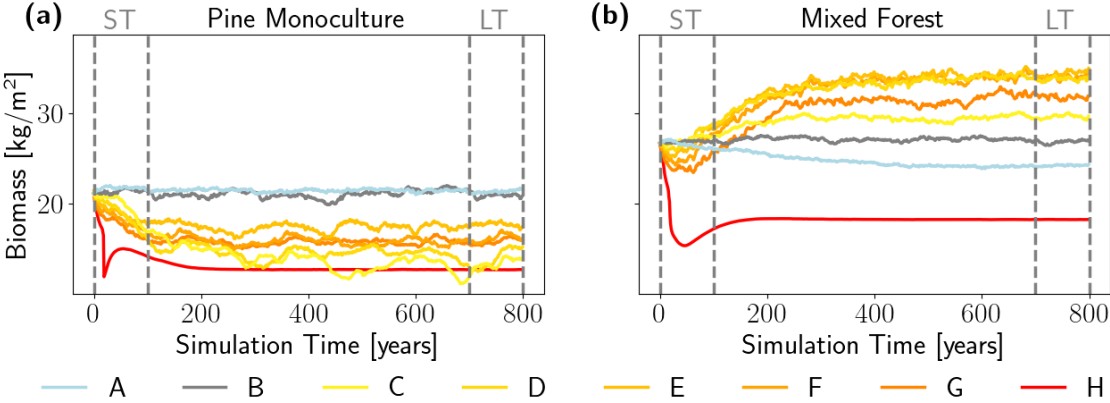


**Figure 2:** *Living biomass in pine monoculture forest (*Pinus sylvestris, *panel (a)) and mixed forest (b) simulated by the*
*LPJmL-FIT DGVM and averaged over Berlin-Brandenburg study area. Living biomass includes above- and belowground*
*biomass [kgC/m²] and was averaged over all patches and grid cells for each year for each drought scenario (Scenarios A-H,*
*see Table 1 for details about the scenarios). Dashed vertical lines mark the limits of the short-term (ST), i.e. the first 100*
*simulation years, and the long-term (LT), i.e. the last 100 simulation years. The wetter-than-the-baseline Scenario A and the*
*baseline Scenario B are shown in blue and grey lines, respectively. The color values of the other points range from yellow to*
*red illustrating the increasing frequency of extreme drought years.*

In the pine monoculture, the amplitude of these fluctuations varied significantly across the different scenarios, and on
average, these fluctuations were much larger than those observed in mixed forests. The standard deviation of the biomass
time series over the last 100 simulated years was, on average, twice as high in the monoculture (0.37 kg/m²) compared to the
mixed forest (0.18 kg/m²), reflecting a stronger response to individual drought years (Fig. 2a).
Both the decrease in biomass for pine monoculture forests and the increase in long-term biomass for mixed forests were non-
linear and non-monotonic, i.e, the variations in biomass levels did not linearly or monotonically correspond to the
differences in drought frequencies or MCWDs among the scenarios (compare to Table 1). The lower drought frequency in
Scenarios C and D resulted in a stronger decrease in biomass compared to the higher drought frequency in Scenarios E, F, G
in the pine monoculture (see Fig. 2a). In the mixed forests, the Scenarios D, E, F have a stronger increase despite a lower
drought frequency than for G and H (see Fig. 2b). Under Scenarios D, E and F, the drought-frequency ranged between 0.28
and 0.53 that resulted in MCWD values of -455, -492 and -521 mm, respectively (Table 1). Surprisingly, despite the large
differences in drought frequency and MCWD, the resulting biomass levels were very close under all three scenarios (Fig.
2b).





**Figure 3:** *Long-term impact of drought frequency on selected forest characteristics under Scenarios A-H as simulated by LPJmL-FIT for the pine monoculture forest (left panels a, c, e, g and i) and the mixed forest (right panels b, d, f, h and j) averaged over the Berlin-Brandenburg study area. Biomass (panel (a) and (b)), Tree Density ((c) and (d)), Height ((e) and (f)), Wood Density ((g) and (h)) and Age ((i) and (j)) are displayed as means over the last simulated 100 years (simulation years 701-800).*





The long-term responses of monocultures and mixed forests to increased drought frequency differed both, at the community
level and at the level of individual trees (Figure 3). At the community level, monoculture vs. mixed forests showed opposing
responses towards increased drought frequency. In the pine monoculture forest, long-term biomass (Fig. 3a) and number of
trees (Fig. 3c) were lower in Scenarios A and B compared to all the drier scenarios (Scenarios C-H). The mixed forest
showed a different pattern. Here, biomass (Fig. 3b) and tree density (Fig. 3d) were higher at the end of the simulation period
the higher the drought frequency became. However, under the extreme Scenario H biomass was lower than Scenario A and
B, while the number of trees was highest (Fig. 3b and d). While height and wood density showed little variation or no trend
across the scenarios for the pine trees growing in monoculture forest (Fig. 3 e and g), increasing drought frequency in the
mixed forest led to decreasing tree height (Fig. 3f) and increased wood density (except under Scenario H, see Fig. 3h). Mean
forest age was lower under Scenarios C-H in the pine monoculture forest compared to Scenarios A and B and showed little
variation (Fig. 3i). On the contrary, trees in the mixed forest grew older the higher the drought frequency became, again with
the exception of Scenario H where average tree age was approx. 50 years lower (Fig. 3j). In general, there was much less
adaptation in individual tree properties and total stand properties in the monocultures compared to the mixed forests. In
mixed forests, trees got smaller, had a higher wood density and grew older with increasing drought frequency while the
monocultures did not show clear trends in the properties of individual trees. It seems that tree demography effects in
conjunction with trait adaptation at the individual level dominate forest adaptation that resulted in the hump-shaped biomass
pattern (Fig. 3b).

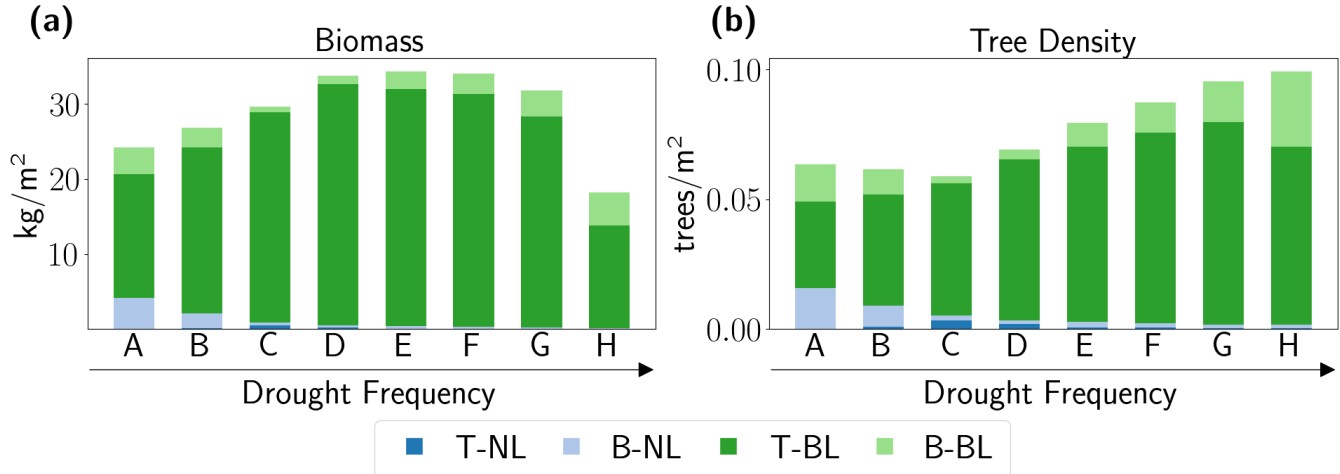


**Figure 4:** *Composition of the plant community in the mixed forests averaged over the Berlin-Brandenburg simulation*
*domain and the last 100 simulation years (701-800). Biomass [kg/m²] (a) and Tree Density [trees/m²] (b) of tree individuals*
*belonging to temperate needle-leaved evergreen PFT (T-NL, dark blue), boreal needle-leaved evergreen PFT (B-NL, light*
*blue), temperate broadleaved summergreen PFT (T-BL, dark green) and boreal broad-leaved summergreen PFT (B-BL,*
*light green) for each drought frequency scenario.*
In addition to changes in biomass and tree density in the mixed forests (as seen in Fig. 3), also their functional composition
shifted in response to increasing drought frequency (Fig. 4). The higher the drought frequency, the lower the proportion of
needle-leaved trees belonging to the T-NL and B-NL PFTs was. Even though needle-leaved trees could still adapt under
Scenarios D and E, their contribution to the overall biomass was marginal in the equilibrium state. Generally, broadleaved
trees dominated the forest community with their biomass being 12 times as high as needle-leaved trees and their tree density
became six times as high as in the baseline Scenario B. While the biomass and tree number of needle-leaved trees further





declined with increasing drought frequency, the number of broadleaved trees increased, as well as their total biomass until
drought frequency was too high (Scenarios F-H). That is, the patterns observed in Figure 3 can be mainly attributed to the
increasing dominance of broadleaved trees. In particular, the temperate broadleaved tree is the dominant PFT in all
scenarios, with its highest fraction in intermediate drought scenarios (Scenarios C-E) and its lowest fraction in the wettest
scenario. In contrast, the fraction of the boreal broadleaved PFT is highest in wettest and driest scenarios, and the boreal
needle-leaved PFT plays only a minor role in the composition, with its highest fraction in Scenario C. Interestingly, the tree
density of boreal broadleaved trees (B-BL) is small under scenarios A and B, is the lowest under Scenario C, and is
increasingly higher under scenarios D-H (Fig. 4b), but does not contribute to biomass at the same proportion (Fig. 4a). Given
those patterns found, it seems that the tree individuals in each PFTs adapt differently to the new climate normals.

**Figure 5:** *Long-term impact of drought frequency on selected tree characteristics. Biomass-weighted mean values are shown*
*for (a) height [m], (b) wood density [kg/m³], (c) age [years] and (d) tree maturity age [years] (also referred to as growth*
*speed, defined as tree height at which a tree reached a height of 15m), specific leaf area [m²/g] (SLA) and mortality*
*[probability/year] at the end of the simulation period (simulation years 701-800) in pine monoculture forest (Pine) and for*
*each PFT in mixed forests under drought frequency Scenarios A-H. Temperate broad-leaved trees (T-BL), boreal broad-*
*leaved trees (B-BL), temperate needle-leaved trees (T-NL) and boreal needle-leaved trees (B-NL). Please note that values*




*for T-NL are based on very low numbers of individuals (see Fig. 4b) and that the sample size for the calculation of tree*
*maturity age is less compared to the data in the other panels, as not all trees reach a height of 15m.*
Figure 4 showed that changes in biomass and tree numbers could mainly be explained by shifts in the functional composition
of mixed forests, i.e. how much biomass and how many trees belonging to a particular PFT contributed to the forest
community. In addition, we observed drought-induced changes in the characteristics of individual trees across PFTs
concerning their height, wood density and age (Fig. 3e-j). In Figure 5, we can show that these changes were rather attributed
to trait adaptation *within* each PFT leading to shifts in mean characteristics of each PFT than to shifts in PFT dominance.The
trends in drought-induced shifts were mostly similar between PFTs for height and wood density but differed for the age of
individual trees. Except for temperate needle-leaved trees (which are very low in number, and therefore need to be treated
with care), the mean height of all PFTs decreased from around 20 m to around 15 m in mixed forests (Fig. 5a). Pine trees
growing in a monoculture only slightly decreased in height, i.e. intra-species plant competition seems to strongly impact
drought adaptation. For wood density (Fig. 5b) we observed differences between broadleaved species, which strongly
increased in density (by 29.6% for T-BL and 19.8% for B-BL), and needle-leaved species, which initially started with a
lower wood density and showed only marginal increases from Scenarios C to H. Pine trees in monocultures showed even
less response than the boreal needle-leaved type. For mean tree age, broadleaved trees generally grew older, while at least
the boreal needle-leaved PFT showed a younger age structure with increasing drought. Again, the results of the temperate
needle-leaved PFT need to be treated with care because of low numbers. In monocultures, the mean age of pine trees was
slightly lower than of their pendant in the mixed forest. Growth speed was generally faster for needle-leaved trees (40-65
years to reach a size of 15 m) than for broadleaved trees (60-85 years). While boreal and temperate broadleaved trees have a
similar growth speed despite their differing wood density, boreal needle-leaved trees seem to grow slower than temperate
ones. In most cases, a higher drought frequency slows down the growth speed, i.e. trees need longer to reach a height of 15
m. However, pine trees in monocultures seem to reach this height faster under mild drought scenarios (Scenarios C and D)
and similar to the baseline scenario for more frequent droughts (Scenarios E-H). For broadleaved trees mortality decreased
with increasing drought frequency and in all drier-than-baseline scenarios (Scenarios C-H) it was lower than for the needle-
leaved trees (Fig. 5f). In contrast, for needle-leaved trees the mortality for the driest scenarios (Scenarios D-H) was higher
than for the baseline scenario. Also for SLA, broadleaved and needle-leaved trees showed opposing trends, although changes
were relatively small for all tree types (Fig. 5e). For the needle-leaved trees SLA increased with drought frequency, while for
the broadleaved trees SLA decreased slightly and for pine there was no significant trend (Fig. 5e). For broad-leaved trees
SLA was higher than for needle-leaved trees, for which in turn SLA was higher than for pine trees and even exceeded the
upper limit of the *Pinus sylvestris* parametrization range.



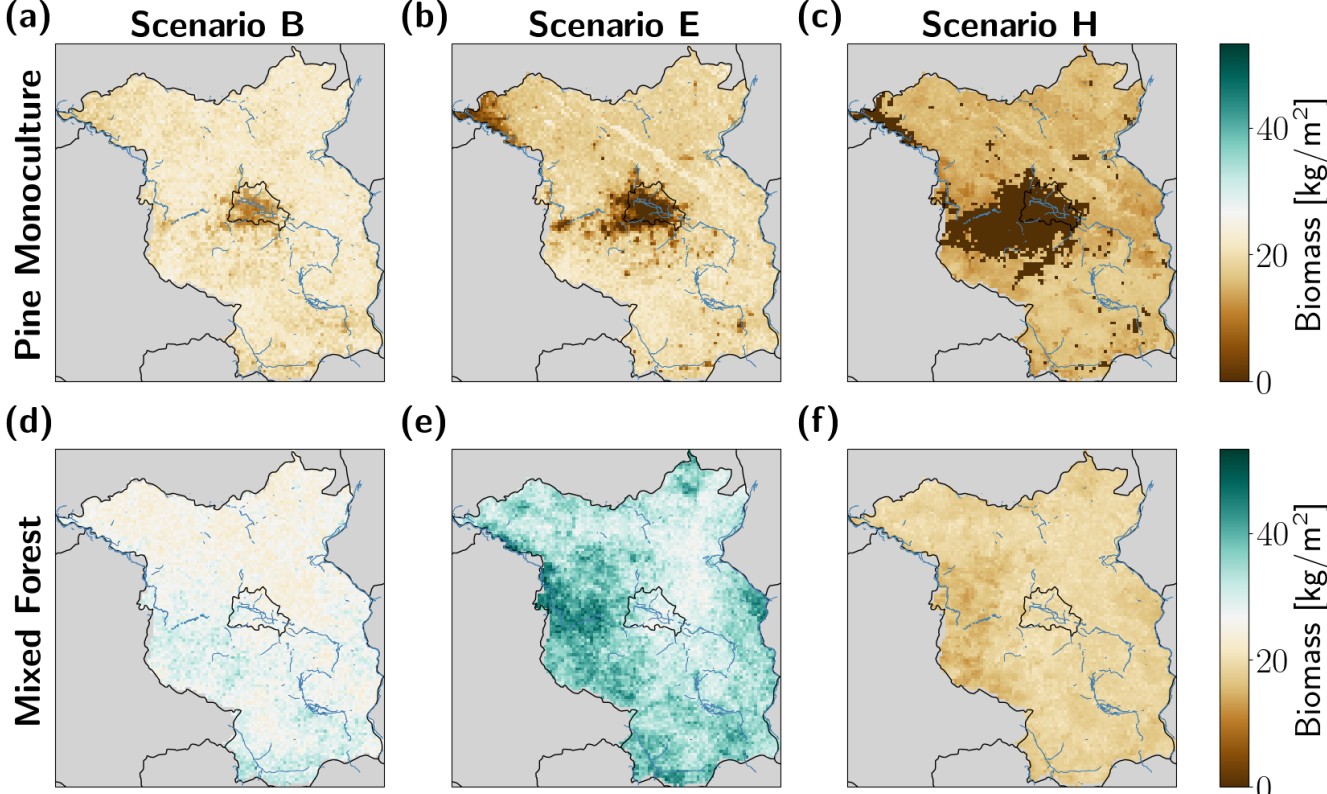


*Figure 6: Simulated long-term (mean over last 100 simulation years) biomass under selected drought frequency scenarios*
*(baseline (Scenario B), example for medium-frequency scenario (Scenario E) and highest drought frequency (Scenario H))*
*for pine monocultures (top row) and mixed forests (bottom row) for the Berlin-Brandenburg study area. The state borders of*
*Berlin and Brandenburg are shown in black, major riverbanks in blue. Biomass in pine monoculture decreased under all*
*scenarios (panels (a) to c)), especially in central and western parts, whereas spatial patterns of biomass increased*
*differently in mixed forests under each scenario (panels (d) to (f)). See Appendix B, Fig B1 for Scenarios A, C, D, F and G.*

The spatial pattern of simulated long-term biomass in the Berlin-Brandenburg study area confirmed the differences between
the performance of pine monocultures and mixed forests under increasing drought frequencies (Fig. 6). A new aspect was,
however, that we saw large spatial heterogeneity for the performance of both, the pine monoculture and the mixed forest,
under baseline climate (Scenario B, Fig. 6a,d). Pine monocultures seemed especially heat sensitive, as Berlin and urban areas
of Brandenburg, which have a higher temperature (see Appendix B, Fig B2), showed much less biomass than in rural areas
(Fig. 6a-c). In contrast, mixed forests seemed to respond to both: temperature increase and water deficits (see Appendix, Fig
B3). Under the baseline scenario, the more or less temporarily stable biomass levels (Fig. 2b) result from biomass values of
around 30 kg/m² in southern Brandenburg and slightly lower biomass values of around 20 kg/m² in northern parts (Fig. 6d).
The higher biomass under medium-frequency Scenario E compared to the baseline (Fig. 2b) which is driven by changes in
structural and functional traits (Figs. 3 and 4) is mainly found in central western and southern parts, and also in isolated areas
in eastern parts of Brandenburg, where biomass reaches values of around 40 kg/m² (Fig. 6e). When the 2018-drought
becomes the new climate normal (Scenario H), lower biomass values (<20 kg/m²) show little spatial variation across the
study area with slightly lower values in western Brandenburg.




Under increased drought frequency (here shown Scenarios E and H), areas of very low biomass in pine monocultures became larger. In the extreme scenario H, they basically collapsed in the western-central Berlin-Brandenburg area and near the river Elbe in north-west Brandenburg (Fig. 6c). As shown before, an increase in drought frequency led to an overall increase in biomass of mixed forests, if drought was not occurring every year (scenario H, Fig. 6e). As a result, biomass in the western-central part of Brandenburg, but not in Berlin was higher in Scenario E, but drastically lower in Scenario H (Fig. 6f).

Summarizing all above results, we observed that mixed forests could better adapt to increasing drought frequency than pine monoculture forests. Mixed forest can adapt via shifts in PFT composition, functional and structural trait shift at the community level which resulted from changes in the characteristics of the tree individuals: The fraction of broadleaved trees increased, and these became sturdier (smaller with higher wood density) and grew older. However there were spatial differences throughout the study areas, with pinus monocultures especially suffering along urban areas and mixed forests responding stronger in the western-central part.

## 4 Discussion

### 4.1 Differences between Pine monocultures and mixed forests under current conditions

Generally, our results for pine monocultures show lower total biomass (around 21 kg/m²) under present conditions (baseline scenario) than mixed forests (around 27 kg/m²). This is in line with a study across forests in northern Germany (Förster et al. 2021) that compared landscape-scale aboveground biomass of different forest types and found much lower values for old-grown pine forests (19 kg/m²) than for forests with natural development (32 kg/m²). Moreover, multiple forest studies show a positive productivity-biodiversity relationship and higher carbon stocks in more diverse forests (Watson et al. 2018; Ali et al. 2017; Y. Zhang and Chen 2015), which can, however, strongly depend on species identity (Chisholm and Dutta Gupta, 2023) and climatic conditions. In a US-scale analysis on forests, Fei et al. (2018) found a positive biodiversity-productivity relationship in arid to semi-arid areas, which turned negative under a more humid climate.

Also the difference in mean tree age between monocultures (118 years) and mixed forests (143 years) is similar to observed values from Förster et al. (2021) who found a mean stand age of 112 years for old-grown pine forests and 146 years for natural developed forests. However, our findings on stem density, which show a slightly higher number of trees per area in mixed forests, deviate from their results, as they find a much higher number of trees in young- and old-grown pine forests than in mixed forests with natural development. This can rather be an effect of management than of tree species composition. However, other studies confirm our results, with higher tree numbers per area in mixed forests than in pine monocultures, e.g. due to a higher use efficiency of light, water, or nutrients by species mixing (Pretzsch and Biber, 2016; Pretzsch and Schütze, 2016). Also our results on slightly higher tree heights and faster growth speed in monocultures are confirmed by other studies (Pretzsch and Forrester, 2017) which is one of the reasons why pine forests have been promoted in forestry of the past decades (Spathelf and Ammer, 2015).

### 4.2 Effects of droughts on biomass and structure of pine monocultures versus mixed forests

Under drier and warmer conditions the difference in total biomass between pine monocultures and mixed forests increases strongly, as the biomass of the monocultures declines further, while the biomass of mixed forests increases in the long-term. Also the stronger fluctuations in the biomass time series show that the pine forest responded stronger to individual drought years, which indicates increased system instability. This is in accordance with observations, which found that pine monocultures were affected by the European drought year 2018, which led to high mortalities of *Pinus sylvestris* and a





vegetation shift to other species (Haberstroh et al., 2022). In addition to the drier conditions, one major reason for the
biomass decline in the pine monoculture were the warmer temperatures negatively affecting the establishment of pine trees
(compare Figure 6 a-c with Table A2), which germinate better with cold stratification (Houšková et al., 2021), and have a
lower thermostability of photosystem II during photosynthesis compared to other important temperate tree species
(Húdoková et al., 2022). However, the model might estimate the chilling temperature needed for cold stratification as too
low for the B-NL PFT and pine trees in Brandenburg, as germination requirements vary with provenance (Hannerz et al.,
2003; Rampart, 2018; Tillman-Sutela and Kauppi, 1995) and might be different for temperate lowlands like Brandenburg
compared to colder regions.
Also broadleaved trees, e.g. European beech *Fagus sylvatica* or even pedunculate oak *Quercus robur,* strongly suffered from
the 2018-drought, but these were less strong and the mixing of appropriate species might have been able to mitigate drought
impacts (Schuldt et al., 2020). In our simulations, we also see that the biomass of mixed forests drops in the first decades, as
they need time to adjust to altered climate conditions. However, after this phase of adjustment, biomass in mixed forests
increases in the long-term (except for the extreme Scenario H). As comparative data on such long-term forest dynamics is
difficult to obtain, it is helpful to assess the validity of our results with the help of a spatial aridity gradient. A comprehensive
study on data from a tree ring database along an aridity gradient in Europe (Pardos et al., 2021) tested the growth response to
site specific drought events between 1975 and 2015 for monospecific and mixed stands. They generally found a higher
resilience and resistance to drought events in mixed forests compared to monocultures but especially an increased resilience
to drought in drier sites with slight increases in productivity in post-drought years for mixed forests but not for monocultures.
Also a long-term study using inventory data of boreal forests of western Canada from 1958 to 2011 confirmed that with
increasing temperature and decreasing water availability, biomass declined in species poor forests while increasing in
species-rich forests (Hisano et al., 2019). These two studies might be an indicator that well adapted species compositions
might indeed become more productive and increase their biomass over centuries in mixed forests. However, the increased
biomass simulated by LPJmL-FIT might be overestimated because the model did not include nutrient limitation, e.g.
nitrogen, which might limit tree growth to a greater extent under drier conditions.
A higher drought frequency does not only impact the biomass of forests, but also their structure. We found large differences
between the pine monoculture and a mixed forest, where different properties of the mixed forest responded much stronger to
increased drought frequency. In the pine forest, tree density decreased in response to more droughts and higher temperatures,
which was also found by Haberstroh et al. (2022). Surprisingly, the pine monoculture does not exhibit the expected shift
towards higher wood density under drought conditions. One explanation lies in the inherent structure of the pinus stands,
which contain a dense population of fast-growing trees with lower wood density under present conditions. These trees,
characterized by rather fast vertical growth, may outcompete new tree saplings with higher wood densities due to light
competition. Therefore, the presence of trees with lower wood densities could hinder the success of slower growing tree
saplings to grow, although trees with higher wood densities might be better adapted to drought conditions. This competition
effect could lead to lower drought resistance of pines growing in a monoculture compared to pines in more diverse forests, as
also found in an empirical study (Granda et al., 2018). Such a phenomenon suggests that light competition in the understorey
plays a significant role in developing certain drought adapted plant strategies. Similar shifts in wood densities as a response
to increased competition has been previously observed in earlier studies (Billing et al., 2024). As the mean height and age
only decreased slightly, the decrease in pine forest biomass seems to be especially caused by lower tree densities and not by
changes in tree morphology.
In contrast, tree density increased in mixed forests, even for the most extreme drought Scenario H. Individual trees, however,
had higher wood density leading to slower growth, reduced mortality, smaller SLA and smaller height, while getting older.
Higher wood density with decreasing annual rainfall (below values of 1000 mm/a, which is the case in our study area) was
also found in a global tree inventory analysis for temperate forests (Bouchard et al., 2024). But also species-specific





investigations showed increasing wood densities for the same species from different provenances across the world with
higher aridity (Nabais et al., 2018). Also Fei et al. (2017) observed that in the Eastern part of the US, tree species whose
range shifted to drier areas had higher median wood density. When trees invest more carbon into their wood density, this can
lead to a decrease in growth rate and size (Aiba and Nakashizuka, 2009; Kallarackal and Ramírez, 2024). Moreover, it makes
sense that trees optimize their height towards smaller sizes, as found in our study, as large trees suffer stronger from droughts
(Bennett et al. 2015, ). Also Ryan and Yoder (1997) found that for the same species, its maximal height can differ strongly
between locations, with smaller tree sizes found in more arid environments. Slower growing forests also have a lower
turnover-rate (Black et al., 2008; Johnson and Abrams, 2009), which is reflected in our finding of an older mean tree age
under higher drought frequency (except for the most extreme Scenario H).

**4.3 Underlying mechanisms leading to a higher resilience of the mixed forest**

As discussed in the previous section, mixed forests seemed to have a higher resilience towards droughts: their biomass
stabilized at high values after an initial adaptation phase, and trees had a higher wood density, grew slower but got older and
reached a lower height. Here, we discuss (i) if the higher resilience is rather a result of a shift in the composition of the
community or in the traits of individual PFTs towards more optimized values, as well as (ii) the general role of species diversity
on forest resilience.
Our results showed a shift in PFT composition in the mixed forest: while needle-leaved trees declined strongly in their
biomass and tree density, playing only a marginal role under increased drought, the biomass and tree density of broadleaved
trees increased. These higher tree densities with increasing aridity have also been observed in a study in northern Germany
on *Fagus sylvatica* monocultural stands (Weigel et al., 2023). Such a community shift from pine trees to broadleaved trees in
response to the drought in 2018 has also been observed in satellite data and tree mortality data in Germany (Haberstroh et al.,
2022). Particularly, temperate broad-leaved trees overall benefited from an increased frequency of the drought-year 2018, as
they are less limited by higher temperatures (Table A2). Looking back at our question if rather the shift in the plant
community composition or a shift in individual tree traits increases the forest resilience of mixed forests, we can say at this
point that a shift towards more temperate broadleaved trees (and the associated shift in community weighted mean traits) can
at least partly explain the higher resilience of mixed forests.
Much more important seems however, the trait shift in individual PFTs as a response to a higher drought frequency. The
flexible-trait scheme of our model allows the emergence of different plant strategies to optimize plant performance under
stressful conditions within a PFT. That is, from our model results, we can learn about PFT-specific plant trait combinations
that are best adapted under different drought conditions. In contrast to needle-leaved species, it seems to be optimal for
broadleaved species to strongly invest into wood density under higher drought frequency. This resembles the well known
coping mechanisms towards high wood densities and slightly smaller SLAs under drought or dry summer conditions
observed at broader scales (Greenwood et al., 2017; Serra-Maluquer et al., 2022) and explains the decrease in mortality
(Greenwood et al. 2017) for broadleaved trees and the overall increase in wood density at the community level. Also in the
study on *Fagus sylvatica* monoculture stands in Germany, lower growth was observed in response to a decadal decrease in
the climatic water balance (Weigel et al., 2023). In general, there seems to be a shift for both broadleaved PFTs towards a
more conservative strategy, where they invest into wood density, grow slower (Chave et al., 2009; Poorter et al., 2010) and
less tall (Aiba and Nakashizuka, 2009; Kallarackal and Ramírez, 2024), but become older (Laurance et al. 2004; Black,
Colbert, and Pederson 2008; Bigler and Veblen 2009). In contrast, boreal needle-leaved trees are less productive, have a
lower wood density thus a higher mortality and consequently the forests are composed of younger and smaller individual
trees with little trait adaptations, which we also see for pine monocultures.





In general, mixed-species forests have been discussed as an adaptation strategy to reduce the risk for forest ecosystems under
global change (Forrester et al., 2016). One reason is the potential niche complementary of different species, reducing
competition for resources (Morin et al., 2011) and improving the resource supply, and uptake (Richards et al., 2010).
Moreover, interspecific facilitation can partly release trees from stress, leading to higher resistance and resilience of mixed
forests especially to climate extremes such as droughts (Pretzsch et al., 2013). However, these findings can strongly depend
on species identity and the environmental context (Decarsin et al., 2024; Forrester et al., 2016). Therefore, the forest species
mixture has to be appropriately chosen for a specific stand to increase the likelihood that beneficiary effects mitigate drought
impacts (Ammer, 2017). As we do not account for these facilitative effects in our simulations, we might even underestimate
the positive effects of mixed forests on drought resilience in our assessments.
In conclusion, it can be summarized that numerous mechanisms lead to the higher resilience via adaptation of mixed forests
to an increased drought frequency, which we have only partially considered in this study. The higher adaptive capacity of
broadleaved trees to drought via shifts in their traits, but also shifts in species composition are both playing a major role.
However, the observation that biomass decreased again when drought frequency was too high shows that also in a mixed
forest the adaptation capacity has limits beyond which productivity decreases.
**4.4 Implications**
We found that an increased drought frequency along with increased mean temperatures adversely affect the productivity of
forests in Berlin-Brandenburg in the first decades. This leads to a massive biomass decline in both forest types, pine
monocultures as well as mixed forests with multiple implications for ecosystem functions and services (for examples see
here: Brockerhoff et al. 2017). However, we also saw that mixed forests can adapt in the long term (as long as extreme
droughts do not occur too often) by a change in species composition towards more broad-leaved trees, but also by shifts in
species traits. Such a shift towards a higher fraction of broad-leaved trees was also suggested in a European-wide study
combining forest inventory data with climate data driven by different Representative Concentration Pathways (RCP)
scenarios which found a retraction in *Pinus sylvestris* and *Picea abies* (Norway spruce) from lowlands in Central Europe to
higher altitudes or more northern areas, but extended areas for *Quercus robur* (pedunculate oak) and ambivalent results for
*Fagus sylvatica* (Buras and Menzel, 2019).
The current and potential future impacts of climate change have concerned practitioners and scientists for more than two
decades (Hanewinkel et al. 2022). However, a case study with forest practitioners in four regions of Germany by Milad et al.
(2013) showed that strategies for adapting forest management were at that point still in early stages. Recently, the Scientific
Advisory Board for Forest Policy of the German Federal Ministry of Food and Agriculture proposed mixed forests to better
cope with climate change, but also the active introduction of better-adapted tree species that are taxonomically, spatially and
ecophysiologically related to current species to also support native biodiversity (Bauhus et al., 2021). Moreover,
using seeds from seed provenances adapted to future climates for reseeding rather than local seed provenances might further
mitigate the initial biomass declines that we found in our simulations, as a recent study on assisted tree migration in Europe
showed (Chakraborty et al., 2024). In order to apply these suggested methods for increasing the resilience towards possible
new climate normals, knowing which traits combinations and species communities perform best under these conditions is
crucial. With our study we contribute to this quest by showing which forest structure and tree characteristics result under
increased drought frequencies in unmanaged forests in Berlin and Brandenburg.
Due to the small grid size of only 2 x 2 km we were able to observe large heterogeneity in the forest biomass across Berlin-
Brandenburg. This heterogeneity increased under increased drought frequencies. However, this could also be a consequence
of the specific spatial heterogeneity in the climate of the year 2018, which has an increasing influence on the results the
higher the frequency becomes. The observed heterogeneity underscores that for the management of forests site-specific



solutions, accounting, e.g. for temperature differences between more urban and more rural areas, for the rainfall gradient in
Berlin-Brandenburg and for different soil textures are required. Furthermore they highlight that in addition to local studies
there is a strong need for high-resolved climate projections that accurately reflect possible increases in extreme drought
frequencies and models that accurately simulate the impacts of these climate projections on vegetation.
**5 Conclusions**
Our results suggest that increased drought frequencies could lead to a drastic reduction in biomass in both pine monoculture
forests and mixed forests in Brandenburg and Berlin. Mixed forests, however, might eventually recover and even exceed
initial biomass levels in the long-term, as long as drought frequencies are not too high. The higher resilience of mixed forests
in our simulations was due to higher adaptive capacity. The adaptation, however, profoundly changed forest characteristics:
Mixed forests were predominantly composed of smaller, broad-leaved trees with higher wood density and slower growth,
which can be summarized as a shift towards more conservative strategies. These changes would have significant
implications for forestry, even when sustainably managed, related industries, and other ecosystem functions and services.
Our results thus highlight the importance of incorporating biodiversity into forest management and preparing for shifts in the
ecosystem services provided by forests in Brandenburg and Berlin in the future.
**Appendices**
**Appendix A: Selected characteristics of plant Functional Types and *Pinus sylvestris* parametrization**
**Table A1:** *Specific leaf area* [m²/g] *(SLA) and wood density* [kg/m³] *(WD) ranges for the simulated plant functional types and* Pinus
sylvestris.

| Plant Functional Type (PFT)/Species | Specific Leaf Area [m²/g] | Wood Density [kg/m³] |
|---|---|---|
| Temperate Broadleaved Summergreen (T-BL) | 0.0242 - 0.0547 | 147.9 - 637.0 |
| Boreal Broadleaved Summergreen (B-BL) | 0.0242 -  0.0547 | 147.9 - 637.0 |
| Temperate Needle-leaved Evergreen (T-NL) | 0.005 - 0.0187 | 117.0 - 418.5 |
| Boreal Needle-leaved Evergreen (B-NL) | 0.005 - 0.0187 | 117.0 - 418.5 |
| *Pinus sylvestris* | 0.0094 - 0.0135 | 223.0 - 268.5 |






**Table A2:** *Temperature limits for tree establishment and survival as well as optimum temperature range for photosynthesis. For establishment to happen, the mean of the annual minimum temperature over the last 20 years must be larger or equal to the Frost Tolerance Temperature [°C] and smaller or equal to the Chilling Requirement Temperature [°C]. Trees die if the mean of the annual minimum temperature over the last 20 years is smaller than the Frost Tolerance Temperature [°C]. The Temperature Optimum for Photosynthesis is the temperature range in which photosynthesis is not inhibited by too low or too high temperatures.*

| Plant Functional Type (PFT)/Species | Chilling Requirement Temperature [°C] | Frost Tolerance Temperature [°C] | Temperature Optimum for Photosynthesis [°C] |
|---|---|---|---|
| Temperate Broadleaved Summergreen (T-BL) | 15.5 | -17.0 | 20.0 - 25.0 |
| Boreal Broadleaved Summergreen (B-BL) | 10 | -35.0 | 15.0 - 25.0 |
| Temperate Needle-leaved Evergreen (T-NL) | 38.0 | -4.0 | 20.0 - 30.0 |
| Boreal Needle-leaved Evergreen (B-NL) | -2.0 | -32.5 | 15.0 - 25.0 |
| *Pinus sylvestris* | -2.0 | -32.5 | 15.0 - 25.0 |

## Appendix B: Biomass, Temperature and Maximum Climatic Water Deficit maps for Berlin-Brandenburg

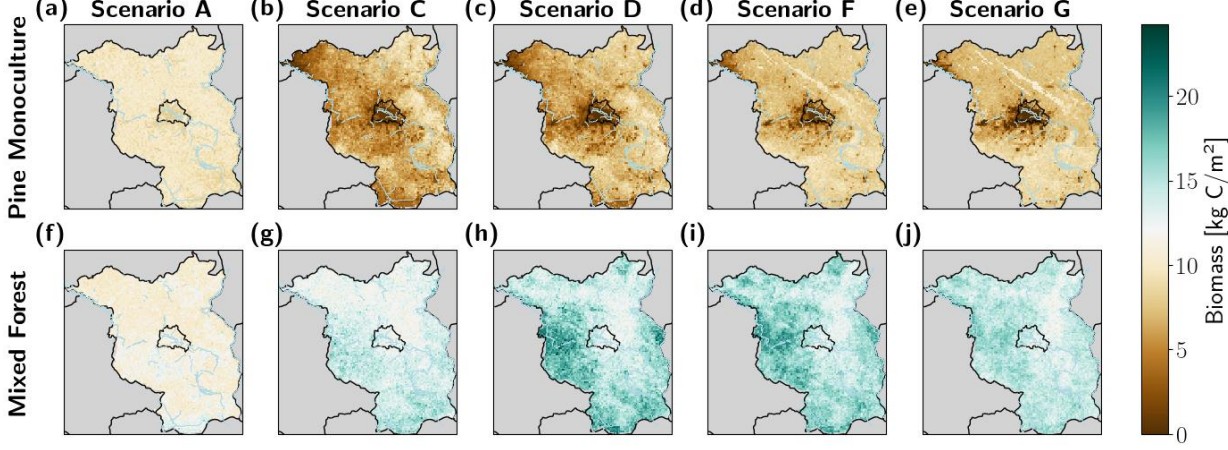



**Figure B1:** *Simulated long-term (mean over last 100 simulation years) biomass under selected drought frequency scenarios A, C, D, F and G for pine monocultures (top row) and mixed forests (bottom row) for the Berlin-Brandenburg study area. The state borders of Berlin and Brandenburg are shown in black, major river banks in blue.*

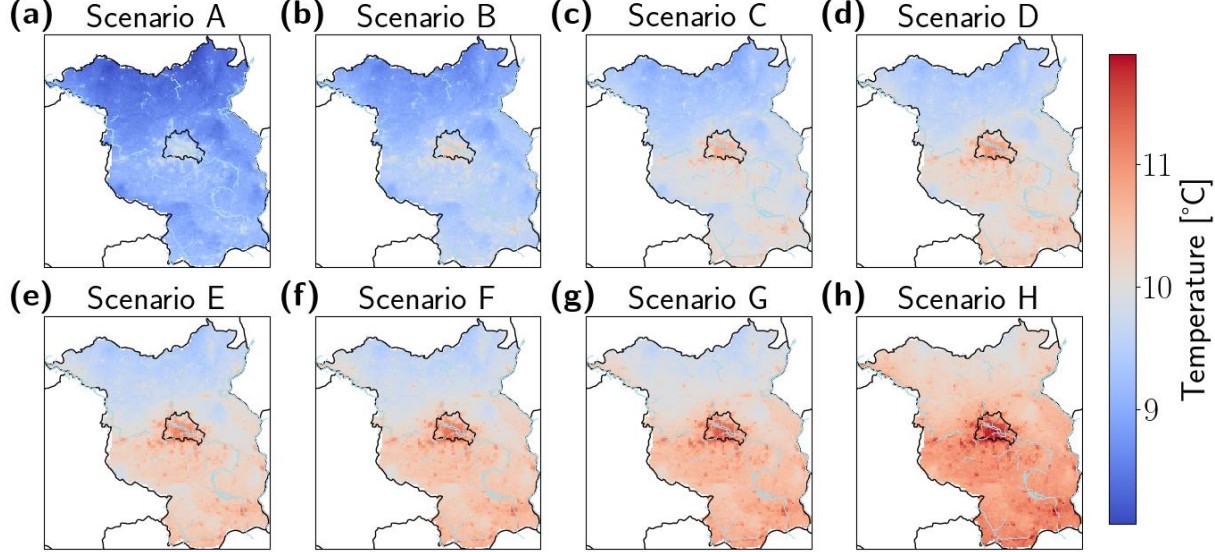

**Figure B2:** *Mean daily Temperatures [°C] over all simulation years for the Berlin-Brandenburg area for scenarios with increasing frequency of the year 2018 from A-H (a-h). The state borders of Berlin and Brandenburg are shown in black, major river banks in blue.*

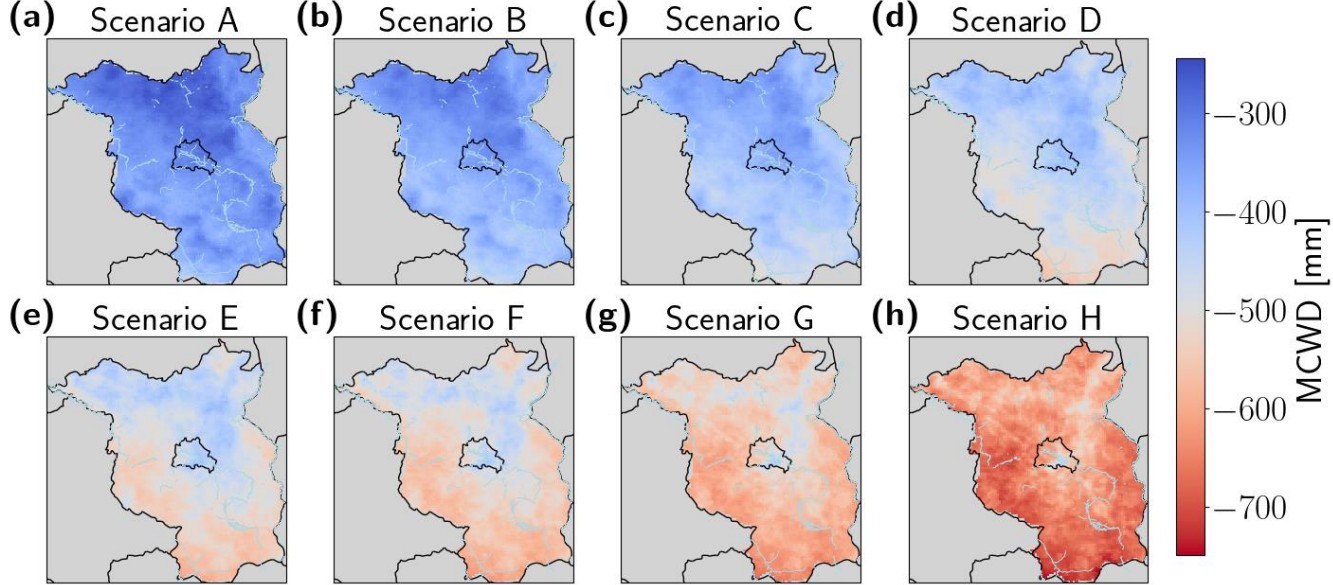



*Figure B3:* *Mean Maximum annual Climatic Water Deficit (MCWD) [mm] over all simulation years for the Berlin-Brandenburg area for scenarios with increasing frequency of the year 2018 from A-H (a-h). The state borders of Berlin and Brandenburg are shown in black, major river banks in blue.*

**Code availability:** Programming code can be provided by Potsdam Institute for Climate Impact Research (PIK e. V.) upon reasonable request and pending scientific review.

**Data availability:** All data needed to evaluate the conclusions in the manuscript are present in the manuscript The data can be provided by Potsdam Institute for Climate Impact Research (PIK e. V.) upon reasonable request and pending scientific review.

**Author contribution:** Jamir Priesner, Britta Tietjen, Kirsten Thonicke, Boris Sakschewski and Maik Billing made the study design. Kirsten Thonicke, Boris Sakschewski, Sarah Bereswill, Werner von Bloh and Maik Billing developed the LPJml-FIT model version used in this study. Jamir Priesner carried out the simulations. Jamir Priesner, Britta Tietjen, Kirsten Thonicke, Boris Sakschewski, Maik Billing and Sebastian Fiedler analyzed and interpreted the simulation outputs. Jamir Priesner, Britta Tietjen and Kirsten Thonicke prepared the manuscript with contributions from all co-authors.

**Competing interests:** The authors declare that they have no conflict of interest.

**Acknowledgements:** This research was funded through the Einstein Research Unit 'Climate and Water under Change' from the Einstein Foundation Berlin and Berlin University Alliance (ERU-2020-609). The authors thank Ainka Douglas for English proofreading the manuscript.

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
