# Peer review of "What if extreme droughts occur more frequently? - Mechanisms and"

_EGUsphere, 2024_

## Author Comment (AC1)

**Reply on RC2**

*This manuscript presents a forest modelling exercise, employing the flexible-individual trait*

*Dynamic Global Vegetation Model LPJmL-FIT, to test the short- and long-term effects of severe drought frequency on biomass, structure and trait distribution of forests in Berlin and Brandenburg states. The manuscript is overall well written and easy to follow for forest ecosystem modelers. The conclusions, though not very novel, are interesting and in general well sustained by model results.*

*In my opinion, more model design/formulation details are needed in the Methods section to appropriately interpret the simulation results. In particular, more details should be given with respect to the impacts of drought on demographic processes in the model (growth, mortality, establishment). Does drought impact biomass accumulation via reduced NPP and growth, or is there an increase in mortality?*

*Do increases in mortality rates result from slow growth or are they directly linked to low soil moisture levels?*

*Does the model include explicit inclusion of plant hydraulics?*

*If these details are not included, the authors expect the reader to be acquainted with LPJmL-FIT design.*

We thank the reviewer for his questions regarding the drought impacts on biomass and mortality in the model as these are important information for the reader to correctly understand and interpret the results.

Currently the model does not have a direct drought-related mortality function, or explicitly includes plant hydraulics. The model also does not contain a direct link between tree mortality and soil moisture to avoid sudden threshold behaviour. Increases in mortality rates ultimately result from slow growth or "growth efficiency", whereas this lower growth efficiency can among other reasons result from water stress/insufficient water supply. Rather "direct" drought effects are captured indirectly in the model: We gave a general description of the model functionality, including how plant productivity, and thus carbon uptake, is connected to transpiration and soil water uptake in the manuscript in lines 155 - 166. Drought can lead to decreased productivity and therefore hamper growth of individual trees leaving a lower amount of carbon to grow new or maintain existing tissues which increases their mortality probability. At the stand scale rather underperforming trees lead to slower accumulation of biomass. If however the overall forest system can somewhat adapt e.g. when trees with higher wood density establish in drier climates, overall mortality is reduced and thus biomass increases (Thonicke et al. 2020). We have now added the following sentence in the Methods section to clearly state that the model considers only indirect effects of drought (via productivity) and currently does not have a drought-mortality function. After line 170 we now write:

"There is no explicit drought-mortality function implemented in the model and also no explicit plant hydraulics. Plant-water uptake and photosynthesis are connected via stomatal conductance. If soil water content is very low, transpiration is reduced affecting photosynthesis and leave phenology which leads to abscission and limits productivity. The reduced productivity results in low growth efficiency and therefore increases mortality (Thonicke et al. 2020)."

We hope that we now provide sufficient information for the reader to understand how drought effects are captured in the model.

*More details on how trait variation, particularly variation in wood density, affects demographic (mortality) rates, should also be given in the Methods section.*

We thank the reviewer for his comment. We add the following lines to the methods section in line 170:

"SLA and WD are key traits to determine the performance of trees under environmental filtering and competition in our model. Trees with high SLA benefit from high production compared to carbon investment, but have higher leaf turnover rates and a lower photosynthetic capacity (Sakschewski et al., 2015). Higher SLA increases the shading of other trees and therefore brings benefits in light competitiveness (Billing et al. 2022). Also WD plays an important role in light competition, as lower WD needs less carbon investment and can lead to faster growth. This can increase productivity and therefore decrease mortality in a light-limited environment. On the other hand, trees with higher WD have a lower probability to die when the annual performance is low, because the maximum of growth-efficiency related mortality is anti-correlated to WD (Sakschewski et al. 2015). For a more in-depth discussion about the trade-offs connected to trait variation see Billing et al (2024), especially Figure 5, and supplementary methods in Billing et al. (2022)."

*It would also be important to state whether SLA and WD values are sampled independently, or they are correlated.*

We thank the reviewer for his helpful remark. SLA and WD are sampled independently. We clarify this in the manuscript by writing "...are then randomly and independently sampled out of the PFT- or species-specific ranges…" in line 153.

*Secondly, the consequences of describing plant diversity using PFTs with respect to known species (except P. sylvestris) is not sufficiently acknowledged in the interpretation of results. To which extent the results can be explained by the fact that needle-leaved PFTs have a narrower trait distribution than broadleaves?*

We thank the reviewer for his question concerning the influence of the width of trait distributions on the results. We assume that the reviewer refers to the range of trait values allowed for PFTs

in the parametrization, shown in Table A1, as we do not show any simulated trait distributions in the manuscript.

We agree that the trait ranges impact the results. Especially in case of SLA, the distribution for needle-leaved PFTs is tilted towards and seems to be cut off at the upper limit of the parametrization range (see Fig. C1). In case of wood density, whose adjustment is crucial for drought adaptation according to our simulations, the distribution is not significantly restricted by the parametrization limits.

We discuss this briefly, starting in line 490:

"Note that the smaller trait ranges from which possible SLA and WD values are drawn for pine trees and needle-leaved PFTs compared to the ranges for broadleaved PFTs by design result in fewer possible trait combinations and therefore fewer possible ways to adapt. However, pine trees do not and the needle-leaved PFTs do only slightly adapt via increased WD and do not use the still open niche space. The long-term mean WD of the pine trees and needle leaved PFTs remains well below the upper limit defined in the parametrization of LPJmL-FIT (see Fig. 5b and Table A1).

In case of SLA of pine trees and needle leaved PFTs the mean of its SLA distribution is rather close to the upper parametrization-limit in all scenarios and is closer to the limit with increasing drought frequency (see Fig. 5c and Table A1). Figure C1 shows exemplarily for scenarios B and H that the SLA distribution for the pine trees and needle leaved PFTs is cut off at the upper limit. That means with increasing drought frequency the environmental and competitive filtering would most likely result in pine trees and needle-leaved PFTs with SLA values higher than those allowed by parametrization and closer to those of the broadleaved trees if there were no limits set. That shows that according to our experiments needle-leaved trees with WD and SLA values in the range of the observations collected in the TRY database for temperate and boreal needle-leaved species do not perform well under scenarios with higher drought frequency."

[Figure]

*Figure C1: Specific Leaf Area (SLA) distribution in the Berlin-Brandenburg study area for pine monoculture (left column) and boreal needle leaved (B-NL) trees growing in a mixed forest (right column) in simulation year 800 of scenario B (upper row) and scenario C (bottom row). Dashed red lines mark the limits of the SLA ranges set in the parametrization.*

*Moreover, PFT trait variability is at least partly a reflect of higher/lower species diversity within the PFT definition. The authors mention adaptive capacity of broadleaved trees, which may be indeed higher than needleleaves, but this may be explained instead by a broader range of trait values in PFT trait distribution arising from a larger number of species included in the PFT definition. Thus, the "adaptation" of mixed forests would rather mean a shift in species identity within the PFT. The low taxonomic resolution of PFTs affects the comparison of the relative importance of composition changes vs trait variation. Therefore, the authors should careful in use terms like "PFT composition" and not "species composition" when discussing results, specially in section 4.3, and in the implications (e.g. L510).*

We thank the reviewer for this constructive comment and agree that we should not use the term "species composition" when referring to model results. We agree with the reviewer that the forest adaptation to drought that is simulated is referring to the shift in tree strategies, which would in reality correspond to species identity. However, because the number of simulated tree strategies emerges from local conditions if translated to species, the taxonomic resolution of PFTs could be high. Because we focus on the trait shifts in the entire forest in this study we cannot make statements on taxonomic or detailed compositional changes. In that sense we follow the suggestion by the reviewer to be more careful in our wording. We went through the whole manuscript carefully and hope that with the committed changes we now distinguish better between PFT composition, trait variability and species composition.

In line 503 and 510 we update the manuscript by writing "PFT composition "instead of "species composition" and in line 511 we now write "PFT traits" instead of "species traits".

Moreover, the reviewer is right that we cannot distinguish between shifts in species identity and shifts in species' traits as we simulate tree strategies that differ in their trait combination and do not resolve individual species. Therefore, we have rewritten our statement and the sentence from line 502 now reads:

"The ability of mixed forests to adapt to increasing drought frequency stems from establishing new, especially broadleaved, trees whose trait combinations have a higher drought tolerance which results in a trait shift."

*Thirdly, if I correctly understood the model design, local filtering applies to adults but it does not imply a change in the trait distribution of future offspring (besides the local extinction of PFTs). In other words, there is no trait inheritability between generations. The authors could discuss whether and how they expect their results to change if this model limitation was removed.*

Thank you for this comment. The reviewer is correct that trees do not pass on traits to the next generation. We agree that trait inheritability between generations could be an interesting way of improving the realism of the model and could impact the results. If included, we expect trait distributions to narrow down and the trait shifts to happen slower when including inheritance of traits. We now briefly discuss this starting in line 515: "In reality, various factors can prevent plant species from occupying all areas that meet their environmental requirements (Lehsten et al., 2019; Lenoir et al., 2020; Thompson and Fronhofer, 2019; Zani et al., 2023), which also depends on the dispersal mechanism, such as seed versus pollen dispersal (Austerlitz and Garnier-Géré, 2003; Cheng et al., 2024; Gerber et al., 2014; Kim et al., 2022). In our model we do not simulate different dispersal strategies. PFTs and trait combinations establish everywhere with the same probability. That means there is no trait inheritance and therefore that the trait combination of new saplings is independent of the previous population's trait distribution. Therefore, especially with respect to trees with local seed dispersal, our simulations might underestimate the time needed for trait shifts and changes in PFT compositions to happen without human intervention."

*Finally, spatial patterns arise from a fine resolution of climate forcing but in my opinion they appear as less interesting for the general reader than other results focusing on the effect of diversity and trait variation, and are a bit disconnected to them. Therefore, they could be omitted or moved to appendix.*

We thank the reviewer for this comment and agree. We now moved this section to the appendix and refer to it in line 291.

"The differences between the performance of pine monocultures and mixed forests under increasing drought frequencies were also confirmed by the spatial pattern of simulated long-term biomass in the Berlin-Brandenburg study area (see Appendix B, Fig. B1). A new aspect was however, that we foundlarge spatial heterogeneity for the performance of both, the

pine monoculture and the mixed forest, under baseline climate (Scenario B, Fig. 6a,d). Pine monocultures seemed especially heat sensitive, as Berlin and urban areas of Brandenburg, which have a higher temperature (see Appendix B, Fig B2), showed much less biomass than in rural areas (Fig. B6a-c). In contrast, mixed forests seemed to respond to both: temperature increase and water deficits (see Appendix, Fig. B3)."

We will delete lines 363-375 from the main text and move them together with Figure 6 to Appendix B, where Figure 6 becomes Figure B1.

*Minor comments:*

*L262: The authors report the following simulation result: The lower drought frequency in Scenarios C and D resulted in a stronger decrease in biomass compared to the higher drought frequency in Scenarios E, F, G in the pine monoculture. I could not find an interpretation of this result, which I found rather surprising.*

We appreciate the referee's observation. Indeed, this is a surprising result. Scenarios C and D not only exhibit lower mean biomass than E, F, and G, despite the lower drought frequency, but also have relatively low mean tree density and a higher variance in mean biomass, mean tree density, mean tree height, and mean age compared to the other scenarios.

To investigate potential reasons for the low mean biomass and density, as well as the high variance across these variables, we examined the climate data used to force the model.We calculated the mean daily precipitation for each month over all years in each scenario. We found that in 2018, mean daily precipitation was higher than in the baseline scenario (scenario B) by 0.074 mm in March and 0.030 mm in April but much lower in summer (-0.77 mm in June, -0.276 mm in July, and -0.784 mm in August). We visualized the mean daily precipitation for each scenario and observed that, since scenarios D–G were derived from scenario C by increasing the frequency of the year 2018, their mean daily precipitation values for each month align between scenarios C and H according to their drought frequency (Figure D1). Specifically, increasing the  drought frequency affects seasonal distribution of precipitation differently, where increased 2018 frequency leads to increased precipitation in March and April for scenarios D-H but increasing frequency of the year 2018 reduces summer precipitation (Figure D1) as one would expect.

This distinct relationship between spring and summer precipitation in scenarios C and D, compared to E–H, may drive different vegetation dynamics. Wetter conditions in March and April favour tree productivity despite the summer precipitation deficit compared to scenarios where spring precipitation remains close to average conditions, but the summer precipitation is still lower than average and impacted by the 2018-like hot and dry conditions. The temperate climate conditions result in light and temperature and - with the increasing frequency of extreme drought years - also water limitation and affects pines of different heights growing in

monoculture. Because the pine monoculture has a smaller WD range and cannot fully shift its niche space due to the combination with the SLA range, reduced productivity (Fig. D2) and hence lower biomass are a consequence of the shift in seasonal precipitation distribution (see also our response above). It also shows the importance of how the different seasons are changed in extreme drought years and how that affects vegetation productivity and related biomass. Because tree mortality increases with increasing drought frequency in scenarios C-G (Fig. 5f) means that the summer precipitation deficit following an average March and April precipitation amount are decisive factors compared to the spring-productivity effect for needle-leaved trees.

We have now added the following sentences starting in line 415: "It is surprising that despite the lower drought frequency in scenarios D and G, simulated biomass is lower than under scenarios with higher drought frequency (scenarios E-H). This can, however, be explained by changes in the seasonal precipitation distribution, where a rather wet April and March in 2018 favoured tree productivity before the hot dry conditions occurred in summer and started to stress the trees (Figure D1). A low 2018-like frequency means more average growing conditions in early spring followed by a hot dry summer, which still reduces biomass. A higher frequency of 2018-like drought conditions means above-average growing conditions in early spring but more severe drought conditions in summer which results in a pine monoculture of slightly smaller and younger trees storing more biomass (see Fig. 5 for details)."

[Figure]

*Figure D1: Mean daily precipitation [mm/day] for each month over all years in each scenario A-H.*

[Figure]

*Figure D2: Net primary production (NPP) in pine monoculture forest (Pinus sylvestris, panel (a)) and mixed forest (b) simulated by the LPJmL-FIT DGVM and averaged over Berlin-Brandenburg study area. NPP was averaged over all patches and grid cells for each year for each drought scenario (Scenarios A-H, see Table 1 for details about the scenarios). Dashed vertical lines mark the limits of the short-term (ST), i.e. the first 100 simulation years, and the long-term (LT), i.e. the last 100 simulation years. The wetter-than-the-baseline Scenario A and the baseline Scenario B are shown in blue and grey lines, respectively.*

*L277-278: Scenarios A and B had "higher", not "lower", biomass and tree density.*
Thanks for spotting this error. We agree and will correct this in the manuscript.

*L439-440: "These trees, characterized by rather fast vertical growth, may outcompete new tree saplings with higher wood densities due to light competition. Therefore, the presence of trees with lower wood densities could hinder the success of slower growing tree saplings to grow, although trees with higher wood densities might be better adapted to drought conditions."*

*I wonder to which degree this result is an artifact of model design. The result may be influenced by the random sampling of traits from species-level distribution, instead of having recruits with trait values taken from distributions influenced by adult trait composition.*

We thank the referee for his remark. We would like to clarify that every time new tree saplings are established in the model, the trait combinations are taken randomly from the observed trait range as derived from the TRY database following the trait-combination rules of the LPJmL-FIT model. It is not a sampling from a species-level distribution. However, one should note that the trait ranges taken from the TRY database were not filtered by age, as this information is not provided for many entries. Therefore, LPJmL-FIT might overestimate possible trait ranges for tree saplings. However, studies focusing on adult trees found wood density values in the same range as the ones resulting in our simulations (Nabais et al., 2018; Torresan et al., 2024). Therefore we do not think that taking the trait range from the trait database to limit trait combinations unfiltered for age-effects causes a strong bias.

*Moreover, wood density may be regarded as a trait varying in time for the same individual, so that all trees in the patch should shift to higher wood density with drought impacts, which would slow vertical growth for all of them.*

We thank the referee for his helpful remark. Indeed, the wood density of trees varies in time due to age, environmental factors and competition (Franceschini et al., 2013; Torresan et al., 2024), which is not considered in our model. However, in the manuscript we proposed that competition for light might dominate the dynamics of the pine monoculture and be the reason why the adaptation towards higher wood density and slower growth seen in the mixed forest does not happen. This principle is independent of whether the wood density can change for individual trees or not. We now write "trees" instead of "tree saplings" in the mentioned paragraph starting in line 436:

"One explanation lies in the inherent structure of the pinus stands, which contain a dense population of fast-growing trees with lower wood density under present conditions. These trees, characterized by rather fast vertical growth, may outcompete trees with higher wood densities due to light competition. Therefore, the presence of trees with lower wood densities could hinder the success of slower growing trees to grow, although trees with higher wood densities might be better adapted to drought conditions. This competition effect could lead to lower drought resistance of pines growing in a monoculture compared to pines in more diverse forests, as also found in an empirical study (Granda et al., 2018). Such a phenomenon suggests that light competition in the understorey plays a significant role in developing certain drought adapted plant strategies. Similar shifts in wood densities as a response to increased competition has been previously observed in earlier studies (Billing et al., 2024). "

**We would like to make further adjustments on our own initiative:**

1. In Figure 3g, 3h, 5b the wood density values are wrongly plotted in units of $kgC/m^3$ (although the label says $kg/m^3$). We now display the values in units of $kg/m^3$.
2. We now acknowledge the funding by the Fachagentur für Nachwachsende Rohstoffe (FNR ) under grant agreement 2219WK39A4.. In the Acknowledgements we now write:

   "This research was funded through the Einstein Research Unit 'Climate and Water under Change' from the Einstein Foundation Berlin and Berlin University Alliance (ERU-2020-609) and by the "Waldspektrum Projekt" funded by the Fachagentur für Nachwachsende Rohstoffe (FNR ) under grant agreement 2219WK39A4."

**References**

Austerlitz, F. and Garnier-Géré, P. H.: Modelling the impact of colonisation on genetic diversity and differentiation of forest trees: interaction of life cycle, pollen flow and seed long-distance dispersal, Heredity, 90, 282–290, https://doi.org/10.1038/sj.hdy.6800243, 2003.

Billing, M., Sakschewski, B., von Bloh, W., Vogel, J., and Thonicke, K.: 'How to adapt

forests?'—Exploring the role of leaf trait diversity for long-term forest biomass under new climate normals, Glob. Change Biol., 30, e17258, https://doi.org/10.1111/gcb.17258, 2024.

Cheng, J., Zhang, M., Yan, X., Zhang, C., Zhang, J., and Luo, Y.: Effects of Seed Size and Frequency on Seed Dispersal and Predation by Small Mammals, Biology, 13, 353, https://doi.org/10.3390/biology13050353, 2024.

Franceschini, T., Longuetaud, F., Bontemps, J.-D., Bouriaud, O., Caritey, B.-D., and Leban, J.-M.: Effect of ring width, cambial age, and climatic variables on the within-ring wood density profile of Norway spruce Picea abies (L.) Karst., Trees, 27, 913–925, https://doi.org/10.1007/s00468-013-0844-6, 2013.

Gerber, S., Chadœuf, J., Gugerli, F., Lascoux, M., Buiteveld, J., Cottrell, J., Dounavi, A., Fineschi, S., Forrest, L. L., Fogelqvist, J., Goicoechea, P. G., Jensen, J. S., Salvini, D., Vendramin, G. G., and Kremer, A.: High Rates of Gene Flow by Pollen and Seed in Oak Populations across Europe, PLOS ONE, 9, e85130, https://doi.org/10.1371/journal.pone.0085130, 2014.

Granda, E., Gazol, A., and Camarero, J. J.: Functional diversity differently shapes growth resilience to drought for co-existing pine species, J. Veg. Sci., 29, 265–275, https://doi.org/10.1111/jvs.12617, 2018.

Kim, M., Lee, S., Lee, S., Yi, K., Kim, H.-S., Chung, S., Chung, J., Kim, H. S., and Yoon, T. K.: Seed Dispersal Models for Natural Regeneration: A Review and Prospects, Forests, 13, 659, https://doi.org/10.3390/f13050659, 2022.

Lehsten, V., Mischurow, M., Lindström, E., Lehsten, D., and Lischke, H.: LPJ-GM 1.0: simulating migration efficiently in a dynamic vegetation model, Geosci. Model Dev., 12, 893–908, https://doi.org/10.5194/gmd-12-893-2019, 2019.

Lenoir, J., Bertrand, R., Comte, L., Bourgeaud, L., Hattab, T., Murienne, J., and Grenouillet, G.: Species better track climate warming in the oceans than on land, Nat. Ecol. Evol., 4, 1044–1059, https://doi.org/10.1038/s41559-020-1198-2, 2020.

Nabais, C., Hansen, J. K., David-Schwartz, R., Klisz, M., López, R., and Rozenberg, P.: The effect of climate on wood density: What provenance trials tell us?, For. Ecol. Manag., 408, 148–156, https://doi.org/10.1016/j.foreco.2017.10.040, 2018.

Sakschewski, B., von Bloh, W., Boit, A., Rammig, A., Kattge, J., Poorter, L., Peñuelas, J., and Thonicke, K.: Leaf and stem economics spectra drive diversity of functional plant traits in a dynamic global vegetation model, Glob. Change Biol., 21, 2711–2725, https://doi.org/10.1111/gcb.12870, 2015.

Thompson, P. L. and Fronhofer, E. A.: The conflict between adaptation and dispersal for maintaining biodiversity in changing environments, Proc. Natl. Acad. Sci., 116, 21061–21067, https://doi.org/10.1073/pnas.1911796116, 2019.

Torresan, C., Hilmers, T., Avdagić, A., Di Giuseppe, E., Klopčič, M., Lévesque, M., Motte, F., Uhl, E., Zlatanov, T., and Pretzsch, H.: Changes in tree-ring wood density of European beech (Fagus sylvatica L.), silver fir (Abies alba Mill.), and Norway spruce (Picea abies (L.) H. Karst.) in European mountain forests between 1901 and 2016, Ann. For. Sci., 81, 49,

https://doi.org/10.1186/s13595-024-01264-5, 2024.

Zani, D., Lischke, H., and Lehsten, V.: Climate and dispersal limitation drive tree species range shifts in post-glacial Europe: results from dynamic simulations, Front. Ecol. Evol., 11, https://doi.org/10.3389/fevo.2023.1321104, 2023.

---

## Author Comment (AC3)

**Reply on RC1**

*This study presents the modelling of forest responses to extreme climate conditions (i.e., droughts), using the dynamic vegetation model LPJmL-FIT. The research focuses on an area in eastern Germany, applying custom climate scenarios to simulate increased probabilities of the 2018 drought event. Two model configurations were used: one representing a monoculture temperate needleleaf evergreen forest and another representing a mixed forest ecosystem with temperate and boreal needleleaf and broadleaf plant functional types.*

*This manuscript presents a valuable test case for the vegetation modeling community by showcasing the capability of vegetation demography models within a scenario-testing framework. Such kind of studies are important because illustrate the critical role in defining optimal forest management strategies under climate change conditions.*

We thank the reviewer for valuing our study. We agree that by applying a scenario-testing framework one can test the impacts of increasing frequency of climate extremes, here drought, on monocultures vs. natural mixed temperate forests.

*However, I believe there are aspects where the manuscript could be improved to better convey its message to the community. Below, I provide a list of suggestions:*

- *Authors focused their attention on the response of mixed forest ecosystems based on the PFT classification. However, reliance on PFTs may inadequately capture the diversity of the different tree species defining the resilience of a forest ecosystem. I think authors should address this issue when introducing/justifying the need of their work and when discussing some of the limitations of their study.*

  We thank the referee for his/her comment. We want to emphasize the fact that we use PFTs with variable traits and that the model simulates different functional tree strategies within a PFT (Thonicke et al., 2020). In our opinion, the diversity of tree species is included in the functional trait space defined by the PFTs. The model approach focuses on capturing functional diversity. We clarify this by writing
  "The potential trait space is defined by these four PFTs and results from all temperate and boreal needle-leaved and broad-leaved trees according to the trait ranges provided in the TRY database. Via environmental and competitive filtering, however, this trait space can be smaller or change (as a result of changing environmental and demographic conditions) but still consists of different tree strategies composing the with-in PFT trait space. The simulated trait space therefore stands for the diversity of all relevant tree species." in line 206.

  *Some recent works (e.g., Forzieri et al., 2022) reported on an emerging signal of declining forest resilience based on trend analysis of remotely sensed information. It would be interesting to apply a similar analysis framework using numerical simulation output presented in this study. In so doing authors would be able to summarize the whole set of model results into more "practical" indicators of ecosystem status.*

We thank the referee for his/her helpful remark. We like his/her idea to use the framework of assessing forest resilience based on temporal autocorrelation of simulated vegetation variables. We will keep it in mind for future work, but it would go beyond the scope of this manuscript.

*The study is based on the pure analysis of numerical simulation results without any level of benchmarking with direct observations and/or forest inventory information. This limitation should be somehow addressed or at least better discussed.*

The model has been validated using observation and remote sensed data products with regard to GPP, biomass, plant height, wood density and SLA (Billing et al., 2022; Thonicke et al., 2020). Due to lack of observation data for the long-term adaptation to droughts in natural temperate forests, no validation could be used for this study specifically. We will discuss that better in the manuscript. Starting in line 233 we now write

"Due to lack of observation data for the long-term adaptation to the occurrence of hot-dry compound events like 2018 in unmanaged temperate forests and from unmanaged temperate forests in general, no benchmarking of our model results with direct observations was possible. Instead we discuss our results qualitatively and where possible also quantitatively referring to the findings of empirical studies from similar environments."

*Minor comments:*

- *I am fine with the creation of ad-hoc drought years in a scenario-testing framework. However, I am not convinced by the arguments used by authors between lines 105-109.*
 *lines 105-109:*
 "However, climate models most likely underestimate the frequency of hot dry compound events like the 2018 drought (Zscheischler and Fischer 2020; van der Wiel et al. 2021) that were much more rare in the past. As a result, vegetation models using these data cannot accurately simulate the impact of increased drought frequency. To overcome this problem, we take a simplistic approach of designing climate scenarios with artificially increased drought frequency for the area of Berlin and Brandenburg in Germany."

We agree that the statement "climate models most likely underestimate the frequency…" might be too strong and will change into "climate models might underestimate the frequency…", which is shown to be true in the both citations given to that statement. Especially the sentence "Thus, projections based on climate model simulations may underestimate the risk of co-occurring hot and dry extremes." in Zscheischler and Fischer 2020 clearly supports our statement. The text now reads:

"However, climate models might underestimate the frequency of hot dry compound events like the 2018 drought (Zscheischler and Fischer 2020; van der Wiel et al. 2021) that were much rarer in the past. Because the realism of the frequency and intensity of such extreme compound events can vary in climate models the resulting simulated impacts on vegetation and tree

demography might be blurred and miss out on possible abrupt changes. Therefore, we take a simplistic approach of designing climate scenarios with artificially increased drought frequency for the area Berlin and Brandenburg in Germany."

- *The color scheme used in Figure 2 makes difficult to easily identify the model response for the different scenarios.*

  We thank the reviewer for this comment and will increase the contrast in colors for the different scenarios.

- *Please check the statement between lines 277-278. It should be the opposite.*
  We agree and will correct the mistake in the manuscript.
- *The first sentence of the Conclusions section does not really apply for mixed forest ecosystems.*
  We thank the reviewer for this comment. Figure 2 shows a reduction in biomass also for mixed forest in the first century of the simulations. In the less severe drought scenarios this reduction in biomass is less than in scenarios with higher drought frequencies. For the scenarios with moderate drought frequency using the expression "drastic reduction in biomass" might be exaggerated and therefore we will simply write "reduction in biomass" instead.

**We would like to make further adjustments on our own initiative:**

1. In Figure 3g, 3h, 5b the wood density values are wrongly plotted in units of kgC/m³ (although the label says kg/m³). We now display the values in units of kg/m³.
2. We now acknowledge the funding by the Fachagentur für Nachwachsende Rohstoffe (FNR ) under grant agreement 2219WK39A4.. In the Acknowledgements we now write:

   "This research was funded through the Einstein Research Unit 'Climate and Water under Change' from the Einstein Foundation Berlin and Berlin University Alliance (ERU-2020-609) and by the "Waldspektrum Projekt" funded by the Fachagentur für Nachwachsende Rohstoffe (FNR ) under grant agreement 2219WK39A4."

**References**

Billing, M., Thonicke, K., Sakschewski, B., von Bloh, W., and Walz, A.: Future tree survival in European forests depends on understorey tree diversity, Sci. Rep., 12, 20750, https://doi.org/10.1038/s41598-022-25319-7, 2022.

Thonicke, K., Billing, M., von Bloh, W., Sakschewski, B., Niinemets, Ü., Peñuelas, J., Cornelissen, J. H. C., Onoda, Y., van Bodegom, P., Schaepman, M. E., Schneider, F. D., and Walz, A.: Simulating functional diversity of European natural forests along climatic gradients, J.

Biogeogr., 47, 1069–1085, https://doi.org/10.1111/jbi.13809, 2020.

van der Wiel, K., Lenderink, G., and de Vries, H.: Physical storylines of future European drought events like 2018 based on ensemble climate modelling, Weather Clim. Extrem., 33, 100350, https://doi.org/10.1016/j.wace.2021.100350, 2021.

Zscheischler, J. and Fischer, E. M.: The record-breaking compound hot and dry 2018 growing season in Germany, Weather Clim. Extrem., 29, 100270, https://doi.org/10.1016/j.wace.2020.100270, 2020.

---

## Author Response (AR2)

**Response to Editor and Reviewer comments:**
**For the next revision please use the initials instead of the full names of the authors for the section "Author`s contribution".**

We thank the Editor for their comment and now we use the initials instead of the full names of the authors for the section "Author`s contribution".

**Editor Márk Somogyvári**

Beside comments from the reviewer, I would ask for the following correction regarding your reply to RC1:

- Please consider mentioning in the conclusions the lack of direct observation data as a potential limitation of the methodology.

We thank Márk Somogyvári for his helpful comments. We now write in line 556 "Another limitation of our methodology is the lack of observation data regarding the long-term adaptation of temperate natural forests to increased frequency of extreme hot-dry compound events and therefore the lack of benchmarking of our model results with direct observations."

Additional private note (visible to authors and reviewers only):
Please check you revised manuscript, to avoid any errors introduced during the review process - as pointed out by the reviewer.

We checked the revised manuscript and corrected punctuation, language and other errors. Most of them were introduced when moving the Figures of the biomass map to the Appendix (now Appendix B, Fig. B1) without changing the references to that Figure in the last round of reviews.

**Suggestions for revision or reasons for rejection, Reviewer 2**

The authors have made a convincing effort to acknowledge model limitations and clarify the interpretation of results.

Please carefully revise the final text, as there might be some unintended mistakes. I point out three that I detected

Caption Fig. 2: "The color values of the other points range from yellow to red illustrating the increasing frequency of extreme drought years." is no longer valid, since you changed the color scale.
L447 (track changes doc): "... in scenarios C and D"
L534: Fig. 5c

We thank reviewer 2 for his helpful comments. We have corrected the mistakes that he pointed out. We removed the sentence "The color values of the other points range from yellow to red illustrating the increasing frequency of extreme drought years." from the caption of Fig. 2.
Instead of "... in scenarios D and G" we now write "... in Scenarios C and D" and instead of "Fig. 5c" now we write "Fig. 5e".